# FINE-GRAINED MACHINE-GENERATED TEXT DETECTION

## ABSTRACT

Machine-Generated Text (MGT) detection identifies whether a given text is human-written or machine-generated. However, this can result in detectors that would flag paraphrased or translated text as machine-generated. Fine-grained classification that separates the different types of machine text is valuable in real-world applications, as different types of MGT convey distinct implications. For example, machine-generated articles are more likely to contain misinformation, whereas paraphrased and translated texts may improve understanding of human-written text. Despite this benefit, existing studies consider this a binary classification task, either overlooking machine-paraphrased and machine-translated text entirely or simply grouping all machine-processed text into one category. To address this shortcoming, this paper provides an in-depth study of fine-grained MGT detection, categorizing input text into four classes: human-written, machine-generated, machine-paraphrased, and machine-translated. A key challenge is the performance drop on out-of-domain texts due to the variability in text generators, especially for translated or paraphrased text. We introduce a RoBERTa-based Mixture of Detectors (RoBERTa-MoD), which leverages multiple domain-optimized detectors for more robust and generalized performance. We offer theoretical proof that our method outperforms a single detector, and experimental findings demonstrate a 5–9% improvement in mean Average Precision (mAP) over prior work on six diverse datasets: GoodNews, VisualNews, WikiText, Essay, WP, and Reuters. Our code and data will be publicly released upon acceptance.

## 1 INTRODUCTION

As Large Language Models (LLMs) have made significant progress in fields like conversational systems (Ouyang et al., 2022; Touvron et al., 2023; Bai et al., 2023), image understanding (OpenAI, 2023; Nori et al., 2023; Ni et al., 2024), and text-to-image generation (Saharia et al., 2022; Rombach et al., 2022; Zhang et al., 2024b; 2023b), concern about the hallucination (Lin et al., 2022) and ethical (Zellers et al., 2019) issues they may raise have increased. To mitigate such misuse, researchers have introduced Machine-Generated Text (MGT) detection to distinguish between human-written and machine-generated text, defending against misinformation. As shown in Figure 1 (A), previous work (Mitchell et al., 2023; Verma et al., 2024; Guo et al., 2023; Zhang et al., 2024a) typically defines this task as a binary classification problem: detecting whether the input text is machine-generated or human-written. However, this binary approach often ignores fine-grained categories of MGT, such as machine-paraphrased or machine-translated text. In practical applications, these fine-grained categories are critical for defending against misinformation and understanding the user intentions of applying LLMs. As illustrated in Figure 1 (B), articles generated by machines based on basic prompts are more likely to contain misinformation (highlighted in pink) or be used for specific purposes (*e.g.*, propaganda or monetization Zellers et al., 2019). In contrast, machine-translated and paraphrased articles modify content based on human-written sources. Users may use LLMs simply to correct grammatical errors in articles. Additionally, providing human-written articles as input increases the cost for bad actors attempting to spread misinformation. While some recent studies (Krishna et al., 2024; Li et al., 2024) have attempted to detect machine-paraphrased text, most still categorize these types as a single class, overlooking the fine-grained differences among these MGT categories. A concurrent study, Abassy et al. (2024), attempts fine-grained MGT detection but addresses solely paraphrased text, ignoring machine-translated text.

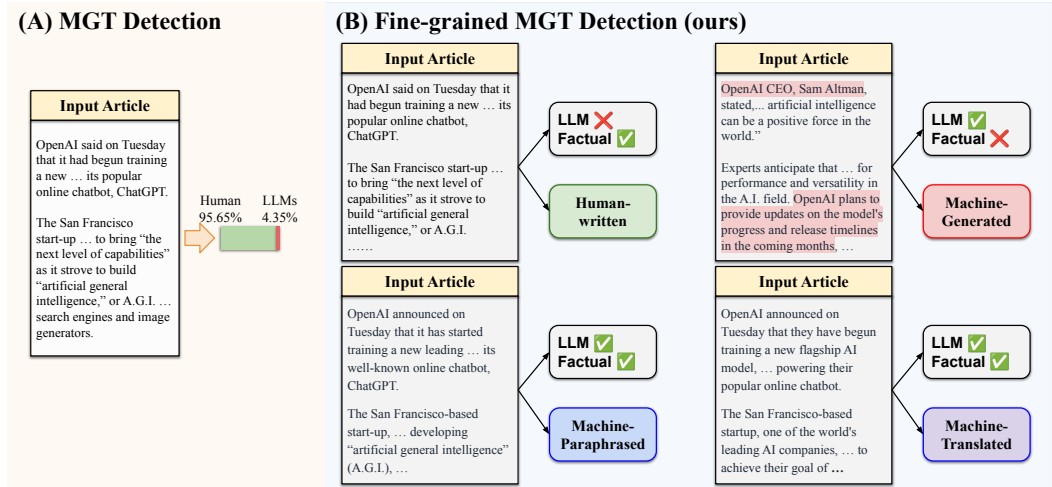

Figure 1: **(A)** Prior work in MGT detection (Mitchell et al., 2023; Su et al., 2023; Guo et al., 2023), predicts a binary label indicating whether the input text is machine- or human-written. Real-world articles are more complex, including human-written text that is machine-paraphrased or machine-translated, which current detectors struggle to identify accurately. We propose RoBERTa-MoD for fine-grained MGT detection, categorizing MGT into three classes: generated, paraphrased, and translated. **(B)** While all three categories of MGT involve LLMs, paraphrased and translated articles are based on human-written sources and do not contain misinformation. In contrast, the article generated from basic prompts includes misinformation, highlighted in pink.

This paper provides an in-depth study of fine-grained MGT detection. Our task classifies a given article into four categories: human-written, machine-generated, machine-paraphrased, and machine-translated. A straightforward method for this task is to modify the classification heads of existing detectors (Solaiman et al., 2019; Guo et al., 2023; Tian et al., 2024) from binary classification to multi-class classification. While this strategy allows us to adapt existing approaches with minimal overhead, these detectors perform poorly on out-of-domain evaluation. As shown in prior work (He et al., 2023; Mitchell et al., 2023; Zhang et al., 2024a), these detectors experience a performance drop on out-of-domain data for binary classification, making fine-grained MGT detection even more challenging.

To solve the aforementioned problems, we propose **RoBERTa**-based **M**ixture **o**f **D**etectors (RoBERTa-MoD) to achieve more robust and generalized MGT detection. Our method employs $M$ detectors, each optimized for a different domain. A gating network is then applied to assign the input article to the most appropriate detectors. With this approach, RoBERTa-MoD can effectively achieve fine-grained MGT detection across various text domains. Experimental results demonstrate that our method outperforms individual RoBERTa-based frameworks, model-averaging ensemble models, and traditional mixture-of-experts ensemble models.

In summary, our contributions are:

- We conduct an in-depth study on fine-grained MGT detection, which is important for identifying misinformation in machine-generated content and understanding the purposes behind users' use of LLMs.
- We introduce a data preparation process to generate articles across different fine-grained categories, enabling the automatic creation of training and evaluation data for our task.
- We identify a key challenge in fine-grained MGT detection: performance degradation in out-of-domain evaluation. To address this, we propose RoBERTa-MoD, combining detectors optimized for different domains to develop a more robust and generalized detection system.
- Our method is validated on six different datasets (GoodNews, VisualNews, WikiText, Essay, WP, and Reuters), achieving a 5∼9 average mAP improvement.

## 2 RELATED WORK

Since LLM-generated articles may contain misinformation (Lin et al., 2022; Zellers et al., 2019) or be used for economic or propaganda purposes (Zhang et al., 2023a), detecting MGT has become increasingly important. Existing methods (Solaiman et al., 2019; Guo et al., 2023; Tian et al., 2024; Mitchell et al., 2023; Hans et al., 2024) typically approach this as a binary classification task, determining whether a segment of text is human-written or machine-generated. Metric-based methods (Mitchell et al., 2023; Su et al., 2023; Bao et al., 2024; Hans et al., 2024) extract distinguishable features from the text using the target language models. *E.g*., Solaiman et al. (2019) apply log probability, and Gehrmann et al. (2019) use the absolute rank of each token. More recently, methods (Mitchell et al., 2023; Su et al., 2023; Bao et al., 2024) have demonstrated that small changes to MGT typically lower its log probability under the language model, a pattern not seen in human-written text. Therefore, these methods introduce perturbations to the input text and measure the resulting discrepancies. To improve the generalization ability of these detectors, Verma et al. (2024) extract features from text using a series of language models and train a classifier to categorize these features. Although these methods do not require additional databases for training, they cannot be easily adapted to fine-grained MGT detection. Since fine-grained categories in MGT are also generated by LLMs, theoretically, machine-translated and machine-paraphrased text would be classified as machine-generated text based on the statistical features extracted by these methods.

Model-based detectors (Solaiman et al., 2019; Guo et al., 2023; Bhattacharjee et al., 2023; Tian et al., 2024; Zhang et al., 2024a) train classifiers on annotated corpora to directly classify input text, making them effective for detecting text generated by black-box or unknown models. *E.g*., Solaiman et al. (2019) finetuned the RoBERTa model (Liu et al., 2019) using outputs from the GPT series. Guo et al. (2023) developed a method to identify ChatGPT-generated text with the HC dataset (Guo et al., 2023). Tian et al. (2024) trained a detector on different scales of text, enhancing the detector's performance on shorter texts. Recently, some studies (Krishna et al., 2024; Li et al., 2024; Nguyen-Son et al., 2021) have recognized the importance of detecting other categories of MGT, including machine-paraphrased and machine-translated text. For example, Krishna et al. (2024) enhanced machine-paraphrased text detection using retrieval methods, and Li et al. (2024) identified paraphrased sentences through the content information in articles. Nguyen-Son et al. (2021) applied round-trip translation to detect Google-translated text. A concurrent study (Abassy et al., 2024) attempted to achieve fine-grained MGT detection. However, it mainly addressed machine-paraphrased text and completely ignored machine-translated text. In our work, we manage to distinguish both machine-paraphrased and machine-translated from MGT. We first modify the classification head of RoBERTa, achieving fine-grained classification on human-written, machine-generated, machine-paraphrased, and machine-translated texts. We further introduce the mixture of detectors to enhance model performance in out-of-domain evaluations, where previous methods have struggled.

## 3 ROBERTA-MOD: ROBERTA-BASED MIXTURE OF DETECTORS

Given an article $\mathbf{x}$, MGT detection uses a binary label $y \in \{\pm1\}$ to classify $\mathbf{x}$ as either human-written or machine-generated. To provide a more precise indication of the article's source, our task further divides MGT into machine-generated, machine-paraphrased, and machine-translated categories, extending MGT detection into a four-class problem where $y \in \{-2, -1, 1, 2\}$.

A straightforward approach is to adapt existing model-based methods (Mitchell et al., 2023; Guo et al., 2023; Tian et al., 2024) by modifying their classification heads from binary to multi-class labels. The primary challenge here is that these models are often trained on specific datasets, leading to decreased performance in out-of-domain evaluations, especially for the more complex fine-grained MGT classification. On the other hand, while metric-based methods (Mitchell et al., 2023; Su et al., 2023; Hans et al., 2024) do not require training on specific data, they typically rely on extracting features from the target LLM and classifying based on predefined thresholds. This approach is not applicable to fine-grained MGT detection since machine-paraphrased and machine-translated texts also contain the statistical characteristics of the target LLM. To address the challenge of out-of-domain evaluation, we propose using mixture models to achieve more generalized and robust performance.

Specifically, Section 3.1 briefly introduces our method for constructing articles used as training and evaluation data. Section 3.2 presents the RoBERTa-based Mixture of Detectors (MoD) strategy. We first initialize multiple detectors through pretraining on corpora, then introduce a routing network to ensemble these detectors and obtain the final score. Furthermore, in Section 3.3, we provide a theoretical proof that for the multi-classification task in fine-grained MGT, mixture models surpass a single detector in performance across various domains.

### 3.1 DATA PREPARATION: ARTICLE GENERATION

As discussed in the Introduction, LLM-generated articles can either be directly produced from basic prompts or be paraphrased or translated based on human-written content. To prepare such data, we generate different MGT categories using article datasets. For the machine-generated category, we provide only the title as the prompt to LLMs, for example: "Write an article on the following title, ensuring that the article consists of approximately $z$ sentences," where $z$ represents the number of sentences in the original article. This ensures that articles of different categories are of similar length, preventing the detector from using length as a classification feature.

For machine-paraphrased and machine-translated articles, we input the entire human-written article as the prompt: "Paraphrase/Translate the following article: $x$." For the translation task, we employed a round-trip translation strategy involving four languages: Chinese, Spanish, Russian, and French. We provide a specific example in Appendix C. The language models used include Llama-3 (Touvron et al., 2023), Qwen-1.5 (Bai et al., 2023), StableLM-2 (Bellagente et al., 2024), ChatGLM-3 (Du et al., 2022), and Qwen-2.5 (Yang et al., 2024)[1]. Llama-3 and Qwen-1.5 are in-domain generators for training the detector, and StableLM-2, ChatGLM-3, and Qwen-2.5 are out-of-domain generators to evaluate the model's generalization ability.

To prevent the model from leaking information about the article's category (*e.g.*, Llama-3 often responds with "Here is the polished version:"), we use the text starting from the second paragraph as input to the detector.

### 3.2 ROBERTA-MOD: ROBERTA-BASED MIXTURE OF DETECTORS

Given an input text $\mathbf{x}$, our model consists a set of $M$ detectors $\{f_1, \ldots, f_M\}$ and a linear gating network $\mathbf{h}$[2]. Denote the parameters of the gating network as $\boldsymbol{\Theta} = [\boldsymbol{\theta_1}, \ldots, \boldsymbol{\theta_M}] \in \mathbb{R}^{d \times M}$, the output of the gating network is $\mathbf{h}(\mathbf{x}; \boldsymbol{\Theta})$, where $d$ is the dimension of the embedded features of $\mathbf{x}$. Denote the output of the $m$-th detector as $f_m(\mathbf{x}; \mathbf{W})$ with input $x$ and parameter $\mathbf{W}$. Note that we simplify the embedded feature $T(\mathbf{x})$ as $\mathbf{x}$ to keep the expression concise, where $T(\cdot)$ is the tokenizer applied in each detector $f_m$ and the gating network $\mathbf{h}$.

The route gate value for $m$-th detector is given by:

$$\pi_m(\mathbf{x}; \boldsymbol{\Theta}) = \frac{\exp(h_m(\mathbf{x}; \boldsymbol{\Theta}))}{\sum_{m'=1}^{M} \exp(h_{m'}(\mathbf{x}; \boldsymbol{\Theta}))}, \forall m \in [M], \tag{1}$$

and the output of MoD is given by:

$$F(\mathbf{x}; \boldsymbol{\Theta}, \mathbf{W}) = \sum_{m \in \mathcal{T}_{\mathbf{x}}} \pi_m(\mathbf{x}; \boldsymbol{\Theta}) f_m(\mathbf{x}; \mathbf{W}), \tag{2}$$

where $\mathcal{T}_{\mathbf{x}} \subseteq [M]$ is a set of selected indices.

**RoBERTa Detector.** Each detector $f_m$ of our method applies the RoBERTa (Liu et al., 2019) architecture. The output of $f_m$ corresponds to four classes: human-written, machine-generated, machine-paraphrased, and machine-translated text.

**Training Strategies.** To develop a method that can be adapted to different detectors, we adopt a two-stage training strategy. First (Figure 2 A), we train detectors separately on various corpora. For the $m$-th detector, the corresponding loss is

---

[1]For Essay, WP, and Reuters, we directly used LLM-generated texts provided by He et al. (2023).

[2]We define the symbols and the data sampling strategy in Section 3.3 following Chen et al. (2022).

Figure 2: **Illustration of RoBERTa-MoD.** For each input $\mathbf{x}$, the router selects top-$k$ detectors to perform predictions according to the output of the router (dotted line). See Section 3.2 for discussion.

**Algorithm 1** Gradient descent for RoBERTa-MoD

**Require:** Number of iterations $T_1$ for $\mathbf{f}$, number of iterations $T_2$ for MoD, learning rate hyperparameters $\eta$ and $\eta_r$.
1: Initialize each entry of $\mathbf{W}^{(0)}$, $\mathbf{\Theta}^{(0)}$ independently.
2: **for** $t = 0, 1, \ldots, T_1 - 1$ **do**
3:     Update $\mathbf{W}^{(t+1)}$ as in 5
4: **end for**
5: **for** $t = 0, 1, \ldots, T_2 - 1$ **do**
6:     Update $\mathbf{W}^{(T_1+t+1)}$ as in 5
7:     Update $\mathbf{\Theta}^{(t+1)}$ as in 6
8: **end for**
9: **return** $(\mathbf{W}^{(T_1+T_2)}, \mathbf{\Theta}^{(T_2)})$.

$$l_m = -\sum_{c=1}^{C} \log \frac{\exp(f_{m,c}(\mathbf{x}; \mathbf{W}))}{\sum_{c'=1}^{C} \exp(f_{m,c'}(\mathbf{x}; \mathbf{W}))} y_{n,c}, \tag{3}$$

$$\mathcal{L}_m = \frac{1}{N} \sum_{n=1}^{N} l_m, \tag{4}$$

where $C$ denotes the number of classes, $N$ denotes the number of samples, and $y_{n,c}$ denotes the target value of $n$-th sample on $c$-th class. In this stage, the parameters $\mathbf{\Theta}$ of the gating network are frozen. We adopted the gradient descent method to update the $\mathbf{W}$ for each detector:

$$\mathbf{W}_m^{(t+1)} = \mathbf{W}_m^{(t)} - \eta \cdot \nabla_{\mathbf{w}_m} \mathcal{L}^{(t)}(\mathbf{\Theta}^{(t)}, \mathbf{W}^{(t)}) / \|\nabla_{\mathbf{w}_m} \mathcal{L}^{(t)}(\mathbf{\Theta}^{(t)}, \mathbf{W}^{(t)})\|_F, \forall m \in [M], \tag{5}$$

where $\eta$ is the detector weight learning rate.

In the second stage (Figure 2 B), we simultaneously update the parameters $\mathbf{W}$ of detectors and the parameters $\mathbf{\Theta}$ of the router. The gradient update rule for $\mathbf{\Theta}$ at iteration $t$ is

$$\theta_m^{(t+1)} = \theta_m^{(t)} - \eta_r \cdot \nabla_{\theta_m} \mathcal{L}^{(t)}(\mathbf{\Theta}^{(t)}, \mathbf{W}^{(t)}), \forall m \in [M], \tag{6}$$

where $\eta_r$ is the learning rate for the router. Algorithm 1 provides the procedure of the training.

### 3.3 RoBERTa-MoD Outperforms Single RoBERTa

Consider a 4-class classification problem over $P$-patch inputs, where each patch has $d$ dimensions. In particular, each labeled data is represented by $(\mathbf{x}, y)$, where input $\mathbf{x} = (\mathbf{x}^{(1)}, \mathbf{x}^{(2)}, \ldots, \mathbf{x}^{(P)}) \in \mathbb{R}^{d \times P}$ is a collection of $P$ patches and $y \in \{\pm 1, \pm 2\}$ is the data label. We consider data generated from $K$ clusters where $k \in [K]$, and for each $k$ has a corresponding feature vector $\mathbf{v_k}$, with $\|\mathbf{v_k}\|_2 = 1$ for $\forall k \in [K]$. For simplicity, we assume that all the vectors $\{\mathbf{v_k}\}_{k \in [K]}$ are orthogonal with each other.

**Definition 1.** *A data pair $(\mathbf{x}, y) \in (\mathbb{R}^{d \times P}, \mathbb{R})$ is generated from the distribution $D$ as follows:*

- *Uniformly draw $k$ and $k'$ from $\{1, \ldots, K\}$ without replacement ($k \neq k'$).*
- *Generate the real data label $y$ and the distracted label $\epsilon$ from $\{\pm 1, \pm 2\}$ uniformly.*
- *Generate two random variables $\alpha, \gamma$ from distribution $D_\alpha, D_\gamma$ independently. In this paper, we assume there exists absolute constants $C_1, C_2$ such that almost surely $0 < C_1 \leq \alpha, \gamma \leq C_2$.*
- *Generate $\mathbf{x}$ as a collection of $P$ patches: $\mathbf{x} = (\mathbf{x}^{(1)}, \mathbf{x}^{(2)}, \ldots, \mathbf{x}^{(P)}) \in \mathbb{R}^{d \times P}$, where*

- *Data Features. One and only one patch is given by $y\alpha\mathbf{v_k}$.*
- *Distracting Features. One and only one patch is given by $\varepsilon\gamma\mathbf{v_{k'}}$.*
- *Gaussian noise. The rest of the $P-2$ patches are Gaussian noises that are independently drawn from $N(0, \sigma_0^2) \cdot \mathbf{I_d}$ where $\sigma_0$ is a variance control constant.*

In Definition 1, the input data $x$ can be decomposed into three components to reflect real-world scenarios: data features that offer relevant information ($y$ and $\mathbf{v_k}$ are closely correlated), distracting features that supply misleading information ($\varepsilon$ and $\mathbf{v_{k'}}$ are randomly selected), and Gaussian noise features introduce some white noise (no useful information) (Chen et al., 2022). For simplicity, we can choose the patch number $P = 3$ without losing generality. Since $\alpha$ and $\gamma$ both serve as scaling parameters for the random-selected features $\mathbf{v_k}$ and $\mathbf{v_{k'}}$, it is safe to assume $\alpha$ and $\gamma$ follow the same distribution $D_\alpha = D_\gamma$.

**Theorem 1.** *(Single detector performs not well). Suppose $D_\alpha = D_\gamma$ holds in Definition 1, then any detector with the form $F(\mathbf{x}) = \sum_{p=1}^P f(\mathbf{x}^{(p)})$ gives poor test performance with the probability $\mathbb{P}_{(\mathbf{x},y)\sim D}(yF(\mathbf{x}) \leq 0) \geq \frac{1}{16}$.*

Theorem 1 indicates that if the distracting feature has the same strength as the data feature i.e., $D_\alpha = D_\gamma$, any two-layer detectors with any activation function cannot perform well on the classification problem defined in Definition 1, with the probability of poor performance being at least $\frac{1}{16}$.

**Theorem 2.** *(MoD performs well). Consider a training dataset of size $n = \Omega(d)$. Let the number of experts $M$ be set to $\Theta(K \log K \log d)$, and the size of the filter $J$ be $\Theta(\log M \log d)$. Under these conditions, the MoD algorithm achieves nearly-zero test error, i.e., $\mathbb{P}_{(\mathbf{x},y)\sim D}(yF(\mathbf{x}; \mathbf{W}) \leq 0) \leq \frac{1}{\beta d}$, where $\beta$ is a constant dependent on the model.*

Theorem 2 demonstrates that MoD could effectively address the multi-classification problem. Linking Theorem 1 and Theorem 2 indicates that under the conditions outlined in Definition 1, the highest error rate of the MoD could be smaller than the lowest error rate of a single-expert model with appropriately selected parameters. This implies that there exist problem instances where an MoD provably surpasses a single-expert model. See Appendix A and B for detailed proof.

## 4 EXPERIMENTS

### 4.1 DATASETS & METRICS

**News Datasets.** The news datasets in our study include *GoodNews* (Biten et al., 2019) and *VisualNews* (Liu et al., 2021). GoodNews(Biten et al., 2019) provides URLs of New York Times articles from 2010 to 2018. After filtering out broken links and non-English articles, we randomly selected 10,000 articles for training, with 2,000 articles each for validation and testing. VisualNews(Liu et al., 2021) comprises articles from four media sources: *Guardian*, *BBC*, *USA Today*, and *Washington Post*. Similar to GoodNews, 2,000 articles were randomly chosen for evaluation sets.

**WikiText** (Stephen et al., 2017) collected 600 training articles, 60 validation articles, and 60 test articles from Wikipedia. We utilize the test set for our evaluation.

**GhostBuster** (Verma et al., 2024) collected corpora for MGT detection from student essays (Essay), creative writing (WP), and news articles (Reuters). In our experiments, we adopt the training, validation, and test sets provided by MGTBench (He et al., 2023), and detect texts generated by various LLMs, including ChatGPT (Ouyang et al., 2022), ChatGLM (Du et al., 2022), GPT4all (Anand et al., 2023), Claude (Anthropic, 2024), and StableLM (Bellagente et al., 2024).

**Metrics.** Following DetectGPT (Mitchell et al., 2023), we use the Area Under the Receiver Operating Characteristic curve (AUROC) to measure performance. We also employ mean Average Precision (mAP) to evaluate performance on articles sampled from specific LLMs. The detector's overall performance is assessed by averaging mAP across various LLMs (avg mAP). To illustrate the method's effectiveness on various fine-grained MGT categories, we utilize confusion matrices for visualization.

Table 1: **Fine-grained MGT Detection on GoodNews.** LLM-DetectAIve is directly trained on fine-grained MGT data, which can be considered as a fine-tuned RoBERTa. "RoBERTa-Avg" denotes averaging the prediction scores from multiple finetuned RoBERTas. "MoE" indicates the application of the traditional Mixture of Experts (Chen et al., 2022) training strategy. RoBERTa-MoD boosts LLM-DetectAIve by approximately 9% in average mAP and 6% in AUROC, demonstrating its effectiveness in detecting fine-grained MGT. See Section 4.3 for detailed discussion.

| Model | In-domain LLMs | | Out-of-domain LLMs | | | **avg mAP** | **AUROC** |
|---|---|---|---|---|---|---|---|
| Scale | Llama3 -8B | Qwen1.5 -7B | StableLM2 -12B | ChatGLM3 -6B | Qwen2.5 -7B | | |
| **mAP on GoodNews (Biten et al., 2019)** | | | | | | | |
| OpenAI-D (base) | 64.95 | 60.25 | 59.51 | 55.04 | 56.74 | 59.30 | 80.06 |
| OpenAI-D (large) | 64.49 | 65.85 | 61.46 | 60.31 | 57.98 | 62.02 | 80.25 |
| ChatGPT-D | 63.85 | 52.76 | 52.65 | 67.18 | 59.62 | 59.21 | 75.41 |
| RoBERTa-MPU | 68.59 | 69.90 | 68.05 | 67.07 | 65.67 | 67.86 | 84.60 |
| LLM-DetectAIve | 87.36 | 79.73 | 77.48 | 76.36 | 72.17 | 78.62 | 89.31 |
| RoBERTa-Avg | 83.28 | 85.64 | 76.48 | 77.50 | 73.49 | 79.28 | 91.02 |
| RoBERTa-MoE | **91.57** | 86.58 | 86.77 | 87.55 | 82.18 | 86.93 | 94.24 |
| RoBERTa-MoD | 91.44 | **91.59** | **87.66** | **87.92** | **82.21** | **88.16** | **95.21** |

## 4.2 BASELINES

**OpenAI-D** (Solaiman et al., 2019) is a detector trained on outputs from GPT-2 (Radford et al., 2019) series. OpenAI provides two versions: RoBERTa-base and RoBERTa-large. With fine-tuning and early stopping, OpenAI-D can also be used to detect text generated by other LLMs.

**ChatGPT-D** (Guo et al., 2023) is designed to identify text produced by ChatGPT-3.5 (Ouyang et al., 2022). It is trained using the HC3 (Guo et al., 2023) dataset, which includes 40,000 questions along with both human-written and ChatGPT-generated answers.

**RoBERTa-MPU** (Tian et al., 2024) builds upon RoBERTa (Liu et al., 2019) by incorporating a length-sensitive loss and a multi-scale text module, addressing the challenges of detecting short texts. Compared to OpenAI-D and ChatGPT-D, RoBERTa-MPU improves the detection for shorter texts without compromising performance on longer texts.

**LLM-DetectAIve** (Abassy et al., 2024) distinguishes between machine-generated, machine-paraphrased, and human-written text by fine-tuning RoBERTa (Liu et al., 2019) and DeBERTa (He et al., 2021) models. For consistency with other baselines, we apply the RoBERTa backbone of LLM-DetectAIve in our experiments.

**Binoculars** (Hans et al., 2024) identifies MGT by comparing the perplexity scores of two pre-trained language models (cross-perplexity), enabling zero-shot detection. Since metrics-based methods classify input text by extracting distinguishable features (*e.g.*, perplexity, absolute rank) from pre-defined LLMs, they are not directly applicable to fine-grained MGT detection. This is because machine-paraphrased and machine-translated texts would still be categorized as machine-generated based on the features in LLMs. Therefore, we only apply this baseline to the traditional MGT detection task.

## 4.3 FINE-GRAINED MGT DETECTION ON GOODNEWS

**Quantitative Results.** Table 1 presents the fine-grained MGT detection results of various models on the GoodNews dataset. All methods were fine-tuned on data from Llama-3 (Touvron et al., 2023) and Qwen-1.5 (Bai et al., 2023), and then evaluated on all LLMs. The maximum token length of the input text was set to 128. We observe that our mixture detectors consistently outperform individual models. For instance, RoBERTa-MoE and RoBERTa-MoD achieve approximately 8.3% and 9.5% improvements in avg mAP compared to LLM-DetectAIve, respectively.

Several conclusions can be drawn from the table. First, the prior knowledge of existing detectors designed for binary classification tasks is not effective for fine-grained MGT detection. For ex-

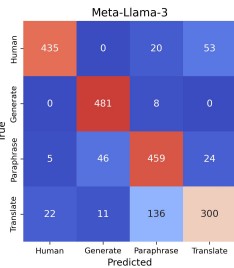 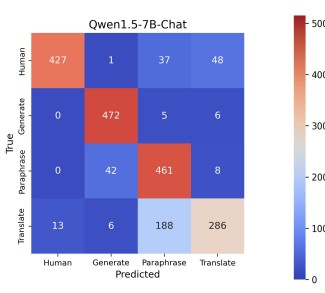

Figure 3: **Confusion Matrix for In-domain Generators.** RoBERTa-MoD performs well in most categories, with the only exception being that machine-translated articles may be misclassified as machine-paraphrased articles. See Section 4.3 for detailed discussion.

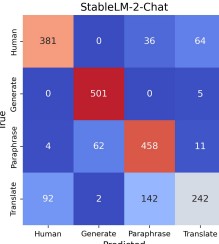 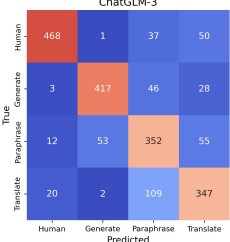 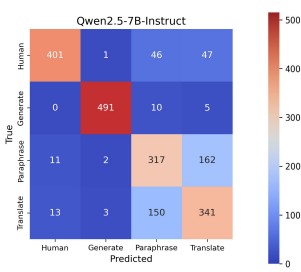

Figure 4: **Confusion Matrix on Out-of-domain Generators.** Our method can still accurately distinguish between human-written and machine-generated categories. Compared to in-domain evaluations, detecting machine-paraphrased and translated text becomes more challenging. See Section 4.3 for detailed discussion.

ample, the performance of RoBERTa-MPU is notably lower than that of LLM-DetectAIve and our RoBERTa-MoD (*e.g.*, 67.86→78.62→88.16 in avg mAP, 84.60→89.31→95.21 in AUROC). Second, mixture models enhance the detection performance of single detectors in both in-domain and out-of-domain evaluations, consistent with our findings in Section 3.3. Third, with the two-stage training strategy, RoBERTa-MoD further boosts the performance of RoBERTa-MoE, demonstrating the effectiveness of using pre-trained detectors to initialize our model.

**Confusion Matrix.** To visualize RoBERTa-MoD's performance across different fine-grained MGT categories, we present the confusion matrices on GoodNews in Figure 3 and 4. The results indicate that RoBERTa-MoD achieves good results in both in-domain and out-of-domain evaluations, particularly for the machine-generated and human-written categories. Distinguishing between machine-translated and paraphrased articles in out-of-domain data (Figure 4) remains more challenging. It may be due to the fact that both machine-paraphrased and translated texts are produced by LLMs using human-written articles as input. Therefore, improving the model's ability to differentiate between these two categories in out-of-domain settings could be a valuable direction for future work.

**Qualitative Results.** Figure 5 presents the qualitative results on GoodNews. We see that the machine-generated article contains significant misinformation, while the translated and paraphrased articles contain fact-based content. This validates the importance of fine-grained MGT detection. Additionally, we observe that the machine-paraphrased and translated articles share similarities in style and content, explaining why the performance for these two categories is less effective than for the human-written and machine-generated categories in Figure 3 and 4.

### 4.4 ZERO-SHOT FINE-GRAINED MGT DETECTION ON VISUALNEWS & WIKITEXT

The experimental results on GoodNews in Section 4.3 show that RoBERTa-MoD outperforms baselines for both in-domain and out-of-domain generators. However, in the same dataset, human-written articles in the training and testing sets may follow similar data distributions. To verify that our model is not overfitting to specific writing styles of GoodNews, we conducted zero-shot exper-

Table 2: **Zero-shot Fine-grained MGT Detection.** Although fine-tuned only on GoodNews articles, Roberta-MoD outperforms LLM-DetectAIve on both VisualNews and WikiText, achieving approximately 5% increases in average mAP. The improvements indicate that RoBERTa-MoD can effectively recognize fine-grained MGT categories without overfitting specific datasets. See Section 4.4 for detailed discussions.

| Model | Llama3 | Qwen1.5 | StableLM2 | ChatGLM3 | Qwen2.5 | **avg mAP** | **AUROC** |
|---|---|---|---|---|---|---|---|
| Scale | -8B | -7B | -12B | -6B | -7B | | |
| **(A) mAP on VisualNews (Liu et al., 2021)** | | | | | | | |
| OpenAI-D (large) | 70.09 | 66.84 | 64.82 | 66.09 | 63.24 | 66.22 | 83.38 |
| ChatGPT-D | 54.91 | 52.73 | 51.40 | 53.45 | 47.50 | 52.01 | 74.23 |
| RoBERTa-MPU | 64.06 | 61.07 | 60.81 | 61.59 | 61.43 | 61.79 | 82.73 |
| LLM-DetectAIve | **78.65** | 70.38 | 67.04 | 65.92 | 66.98 | 69.79 | 84.08 |
| RoBERTa-MoD (Ours) | 73.04 | **81.67** | **71.88** | **72.44** | **71.75** | **74.16** | **89.72** |
| **(B) mAP on WikiText (Stephen et al., 2017)** | | | | | | | |
| OpenAI-D (large) | 64.14 | 66.76 | 58.86 | 51.58 | 57.10 | 59.69 | 76.19 |
| RoBERTa-MPU | 71.14 | 70.64 | 64.67 | 62.52 | 66.61 | 67.12 | 83.01 |
| LLM-DetectAIve | 78.01 | 76.67 | 72.26 | **70.16** | 72.07 | 73.83 | 82.82 |
| RoBERTa-MoD (Ours) | **79.29** | **80.70** | **75.21** | 66.96 | **77.49** | **75.93** | **87.55** |

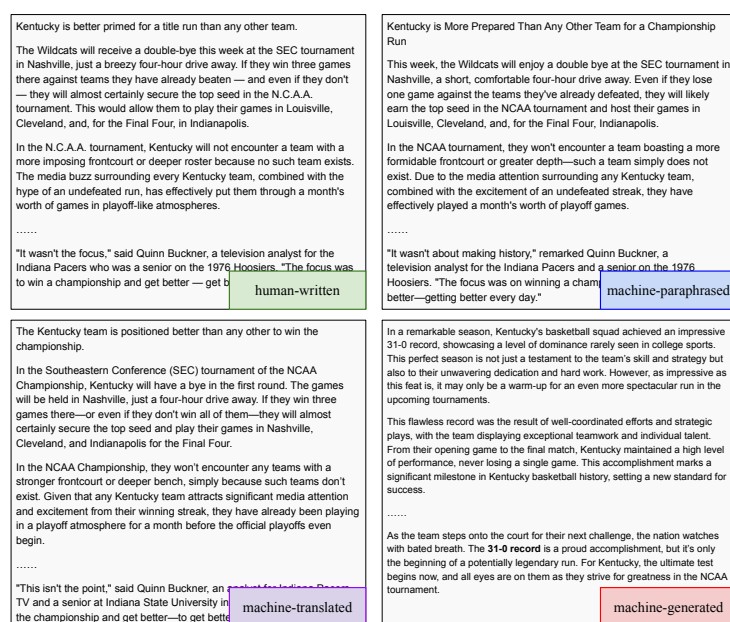

Figure 5: **Qualitative Results on GoodNews.** RoBERTa-MoD effectively classifies articles based on their content. Since both machine-translated and paraphrased texts are generated by LLMs based on human-written sources, they share similar style and content, posing challenges for the detector. The observation is consistent with the results in Figure 3 and 4. See Section 4.3 for discussion.

iments on VisualNews and WikiText, as shown in Table 2. In these evaluations, RoBERTa-MoD continues to outperform the baseline models in both average mAP and AUROC (*e.g.*, 69.79→74.16 in average mAP and 84.08→89.72 in AUROC on VisualNews). This indicates that the improvements made by RoBERTa-MoD are due to its effectiveness in identifying fine-grained MGT, rather than remembering specific human-written styles of GoodNews articles.

## 4.5 MGT DETECTION

Given the flexibility of the MoD strategy, we believe that RoBERTa-MoD can be used not only for fine-grained MGT detection but also for traditional binary MGT classification. To validate this,

Table 3: **MGT Detection on Essay, Reuters, and WP.** RoBERTa-MoD outperforms all RoBERTa-based baselines, including OpenAI-D (Solaiman et al., 2019), ChatGPT-D (Guo et al., 2023), and Roberta-MPU (Tian et al., 2024), and performs comparably to the state-of-the-art, Binocualrs (Hans et al., 2024). See Section 4.5 for a detailed discussion.

| | mAP | | | F1 | | | AUROC | | |
|---|---|---|---|---|---|---|---|---|---|
| | Essay | Reuters | WP | Essay | Reuters | WP | Essay | Reuters | WP |
| OpenAI-D (large) | 78.52 | 92.52 | 83.33 | 63.76 | 77.15 | 65.72 | 77.85 | 93.21 | 83.00 |
| ChatGPT-D | 78.02 | 84.77 | 73.84 | 64.71 | 71.66 | 56.50 | 72.35 | 81.19 | 72.66 |
| RoBERTa-MPU | 89.71 | 97.23 | 96.53 | 73.73 | 91.13 | 79.57 | 87.15 | 96.53 | 96.05 |
| Binoculars | **99.11** | 98.36 | **98.72** | **88.50** | 77.33 | **88.05** | **98.72** | 97.95 | **98.44** |
| RoBERTa-MoD (Ours) | 92.72 | **98.43** | 98.18 | 81.71 | **92.37** | 87.28 | 90.59 | **98.22** | 97.92 |

we conducted binary MGT detection on Essay, WP, and Reuters (Verma et al., 2024), as shown in Table 3. RoBERTa-MoD outperforms model-based baselines (*e.g.*, OpenAI-D, ChatGPT-D, RoBERTa-MPU), demonstrating the effectiveness of our MoD strategy.

While the state-of-the-art method, Binoculars (Hans et al., 2024), performs slightly better than our method, it is worth noting that Binoculars is metric-based and distinguishes between human-written and machine-generated text using a fixed threshold. Therefore, it is not applicable to fine-grained MGT detection, since both paraphrased and translated texts are also generated by LLMs and would exhibit similar metric scores extracted from the target LLMs. In contrast, our method does not rely on these metrics, enabling it to perform well in both binary MGT classification and fine-grained MGT detection tasks.

## 5 LIMITATIONS

In this paper, we highlight the importance of fine-grained MGT classification and identify out-of-domain evaluations (*e.g.*, out-of-domain generators and zero-shot articles) as a primary challenge for this task. We introduced RoBERTa-MoD to improve the performance of existing detectors. Despite improvements across various datasets, our method still has several limitations.

First, out-of-domain evaluations remain a challenge for further improvement. As shown in our experiments, the detectors' performance on out-of-domain generators (StableLM-2, ChatGLM-3, Qwen-2.5) is still lower than that on in-domain generators (Llama-3, Qwen-1.5). Performance in zero-shot experiments (VisualNews and WikiText) is also lower compared to GoodNews.

Short text detection is another issue that could be addressed in future work. In this paper, we set the maximum token length to 128 and achieved reasonable results. However, in our experiments, when the maximum token length is reduced to 32 or lower, almost all models, including RoBERTa-MPU, which is specifically designed for short text detection, perform close to random guessing. Therefore, addressing the short text detection problem in fine-grained MGT detection is a potential direction for future research.

## 6 CONCLUSION

We conduct an in-depth study of fine-grained MGT detection, aiming to further distinguish between machine-paraphrased and machine-translated text from MGT. We identify a key challenge in fine-grained MGT detection as improving the model's generalization ability. *I.e.*, model-based detectors typically perform well on in-domain data, however, their performance declines when dealing with different domains, especially out-of-domain data. To address this challenge, we introduce RoBERTa-MoD, which consists of multiple detectors optimized for different domains, achieving more robust and generalized results in multi-domain evaluations. Our method is evaluated on six datasets (GoodNews, VisualNews, WikiText, Essay, WP, and Reuters), achieving a 5–9% improvement in average mAP compared to baselines. The improvements across various datasets and generators demonstrate the effectiveness of our approach in fine-grained MGT detection.

## ETHICS STATEMENT

In our study, we introduce RoBERTa-MoD to enable fine-grained classification of MGT, which can help prevent the spread of misinformation and identifying the intent behind users' use of LLMs. However, like other methods designed for MGT detection, our system cannot guarantee 100% accuracy, especially in the more challenging fine-grained detection task. While the proposed MoD strategy improves performance in out-of-domain generators and zero-shot evaluations, challenges remain in identifying specific fine-grained categories (as discussed in Section 4.3). Therefore, we strongly discourage the use of our methods without human supervision (*e.g.*, in plagiarism detection or similar scenarios). A more appropriate application of RoBERTa-MoD would be in defending against LLM-generated misinformation under human supervision. Through this paper, we aim to highlight the importance of fine-grained MGT detectors for better distinguishing articles containing misinformation and fact-based articles polished by LLMs.

## REPRODUCIBILITY STATEMENT

Our model is mainly implemented based on Pytorch (Paszke et al., 2019) and Transformers (Wolf et al., 2020). During training, the maximum token length of the input text is set to 512. We limit the maximum length to 128 to evaluate the model's performance in shorter text detection in the test stage. For RoBERTa-MoD, we use a batch size of 16 and a maximum learning rate of $10^{-5}$. We fine-tuned the model for three epochs with an early stopping strategy, following Zhang et al. (2024a); Verma et al. (2024) to prevent overfitting. Our experiments were conducted on RTX-A6000 and other 48GB memory GPUs (*e.g.*, A40, L40S). For a single dataset (*e.g.*, GoodNews), data preparation takes approximately 60 hours, and training takes around 1 hour. We will also release our code upon acceptance to ensure reproducibility.

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

## APPENDIX

## A  PROOF OF THEOREM 1

Followed by Definition 1, the input features $\mathbf{x}$ consists of data features, distracting features, and Gaussian noise features, so it can be expressed as $\mathbf{x} = [\alpha y \mathbf{v}_k, -\gamma y \mathbf{v}_{k'}, \boldsymbol{\xi}]$ where $\boldsymbol{\xi}$ is the Gaussian noise vector. For simplicity, we choose the patch number $P = 3$ without losing generality.

We assume that $\gamma$ and $\alpha$ are identically distributed, such that $D_\alpha = D_\gamma$. Given that both $y$ and $-y$ belong to the set $\{\pm 1, \pm 2\}$, it results that $y$ and $-y$ follow the same distribution. Conditioned on the event that $y = -\epsilon$, points $([\alpha y \mathbf{v}_k, -\gamma y \mathbf{v}_{k'}, \boldsymbol{\xi}], y)$, $([-\alpha y \mathbf{v}_k, \gamma y \mathbf{v}_{k'}, \boldsymbol{\xi}], -y), ([\gamma y \mathbf{v}_{k'}, -\alpha y \mathbf{v}_k, \boldsymbol{\xi}], y), ([-\gamma y \mathbf{v}_{k'}, \alpha y \mathbf{v}_k, \boldsymbol{\xi}], -y)$ follow the same distribution. Therefore, we can express the conditional probability $\mathbb{P}(yF(\mathbf{x}) \leq 0 \mid \epsilon = -y)$ as

$$\begin{aligned} &4\mathbb{P}(yF(\mathbf{x}) \leq 0 \mid \epsilon = -y) \\ &= \mathbb{E}[\mathbb{1}\left(yF\left([\alpha y \mathbf{v}_k, -\gamma y \mathbf{v}_{k'}, \boldsymbol{\xi}]\right) \leq 0\right) + \mathbb{1}\left(-yF\left([-\alpha y \mathbf{v}_k, \gamma y \mathbf{v}_{k'}, \boldsymbol{\xi}]\right) \leq 0\right) \\ &\quad + \mathbb{1}\left(yF\left([\gamma y \mathbf{v}_{k'}, -\alpha y \mathbf{v}_k, \xi]\right) \leq 0\right) + \mathbb{1}\left(-yF\left([-\gamma y \mathbf{v}_{k'}, \alpha y \mathbf{v}_k, \boldsymbol{\xi}]\right) \leq 0\right)]. \end{aligned} \tag{7}$$

Apply function $f(\cdot)$ to each patch of $\mathbf{x}$ to obtain the following result

$$\begin{aligned} &(yF\left([\alpha y \mathbf{v}_k, -\gamma y \mathbf{v}_{k'}, \boldsymbol{\xi}]\right)) + (-yF\left([-\alpha y \mathbf{v}_k, \gamma y \mathbf{v}_{k'}, \boldsymbol{\xi}]\right)) \\ &+ (yF\left([\gamma y \mathbf{v}_{k'}, -\alpha y \mathbf{v}_k, \boldsymbol{\xi}]\right)) + (-yF\left([-\gamma y \mathbf{v}_{k'}, \alpha y \mathbf{v}_k, \boldsymbol{\xi}]\right)) \\ &= (yf\left(\alpha y \mathbf{v}_k\right) + yf\left(-\gamma y \mathbf{v}_{k'}\right) + yf\left(\boldsymbol{\xi}\right)) + (-yf\left(-\alpha y \mathbf{v}_k\right) - yf\left(\gamma y \mathbf{v}_{k'}\right) - yf\left(\boldsymbol{\xi}\right)) \\ &+ (yf\left(\gamma y \mathbf{v}_{k'}\right) + yf\left(-\alpha y \mathbf{v}_k\right) + yf\left(\boldsymbol{\xi}\right)) + (-yf\left(-\gamma y \mathbf{v}_{k'}\right) - yf\left(\alpha y \mathbf{v}_k\right) - yf\left(\boldsymbol{\xi}\right)) \\ &= 0. \end{aligned} \tag{8}$$

Given that an input feature $\mathbf{x}$ with all-zero patches is practically meaningless, we will exclude this scenario from consideration. In such cases, at least one identical function $\mathbb{1}(\cdot)$ in Eq. 7 will be non-zero. This implies that

$$4\mathbb{P}(yF(\mathbf{x}) \leq 0 \mid \epsilon = -y) \geq 1. \tag{9}$$

Applying $\mathbb{P}(\epsilon = -y) = 1/4$ and the Bayes' rule, we have that

$$\mathbb{P}(yF(\mathbf{x}) \leq 0) = \mathbb{P}(yF(\mathbf{x}) \leq 0) \mid \epsilon = -y)\mathbb{P}(\epsilon = -y) \geq 1/16. \tag{10}$$

## B  PROOF OF THEOREM 2

Drawing inspiration from the proof strategy in Lemma 5.2 by Chen et al. (2022), we focus on the $m$-th expert in the MoE layer, assuming that $m \in \mathcal{M}_k$. The bounds for the inner product between the weights and the freshly drawn i.i.d random noise from $\mathcal{N}\left(0, (\frac{\sigma_p}{\sqrt{d}})^2 \cdot \mathbf{I}_d\right)$ is necessary. Let $\frac{\sigma_p}{\sqrt{d}} = \sigma_0$

for convenience. Normalized gradient descent with a step size of $\eta$ is adopted in the updating stage, we can have

$$\left\| \mathbf{w}_{m,j}^{(T)} - \mathbf{w}_{m,j}^{(0)} \right\|_2 \leq \eta T = \widetilde{O}(1). \tag{11}$$

Using the triangle inequality on Eq. 11, we derive that

$$\left\| \mathbf{w}_{m,j}^{(T)} \right\|_2 \leq \left\| \mathbf{w}_{m,j}^{(0)} \right\|_2 + \widetilde{O}(1). \tag{12}$$

Furthermore, the inner product $\left\langle \mathbf{w}_{m,j}^{(t)}, \boldsymbol{\xi} \right\rangle$ adheres to the distribution $\mathcal{N}\left(0, (\sigma_0)^2 \cdot \left\| \mathbf{w}_{m,j}^{(T)} \right\|_2^2\right)$, with probability at least $1 - \frac{1}{dPMJ}$. Define $\beta$ as a model-related parameter proportional to $P, M, J$. If the MoD model is fixed, $\beta$ remains constant. We can have

$$\left| \left\langle \mathbf{w}_{m,j}^{(T)}, \boldsymbol{\xi} \right\rangle \right| = O\left( \sigma_p d^{-1/2} \left\| \mathbf{w}_{m,j}^{(t)} \right\|_2 \log(dPMJ) \right) \leq \widetilde{O}(\sigma_0). \tag{13}$$

Applying Boole's inequality for $m \in [M], j \in [J]$ gives that, with probability at least $1 - \frac{1}{\beta d}$,

$$\left| \left\langle \mathbf{w}_{m,j}^{(T)}, \boldsymbol{\xi} \right\rangle \right| = \widetilde{O}(\sigma_0), \forall m \in [M], j \in [J]. \tag{14}$$

Expanding the inner product in Eq. 14, we have that

$$\begin{aligned} yf\left(\mathbf{x}, \mathbf{W}^{(T)}\right) &= y \sum_{j \in [J]} \sum_{p \in [P]} \sigma\left(\left\langle \mathbf{w}_{m,j}^{(T)}, \mathbf{x}^{(p)} \right\rangle\right) \\ &= y\sigma\left(\left\langle \mathbf{w}_{m,j}^{(T)}, \alpha y \mathbf{v}_k \right\rangle\right) + y \sum_{(j',p) \neq (j,1)} \sigma\left(\left\langle \mathbf{w}_{m,j'}^{(T)}, \mathbf{x}^{(p)} \right\rangle\right). \end{aligned} \tag{15}$$

Incorporating the inequality in Lemma E.12 from Chen et al. (2022), we can get

$$yf\left(\mathbf{x}, \mathbf{W}^{(T)}\right) \geq C_1^3 \left(1 - \sigma_0^{0.1}\right)^3 M^{-4} - \widetilde{O}(\sigma_0^3) = \widetilde{\Omega}(1) \geq 0. \tag{16}$$

Because Eq. 14 holds with probability at least $1 - \frac{1}{\beta d}$, we can have

$$\mathbb{P}_{(\mathbf{x},y)\sim\mathcal{D}}\left(yf\left(\mathbf{x}; \mathbf{W}^{(T)}\right) \geq 0\right) \geq 1 - \frac{1}{\beta d}, \tag{17}$$

which is equivalent to

$$\mathbb{P}_{(\mathbf{x},y)\sim\mathcal{D}}\left(yf\left(\mathbf{x}; \mathbf{W}^{(T)}\right) \leq 0\right) \leq \frac{1}{\beta d}. \tag{18}$$

## C  ROUND-TRIP TRANSLATION STRATEGY

As discussed in Section 3.1, we adopt the strategy of round-trip translation to generate translation data for fine-grained MGT detection. Figure 6 provides a specific example: we first translate the original article into target languages (Chinese, Spanish, French, Russian), and then translate these articles back into English, obtaining machine-translated articles for detection.

## D  FINE-GRAINED MGT DETECTION WITH DIFFERENT INPUT LENGTHS

We report the performance of RoBERTa-MoD with different input lengths in Table 4. We observe that as the input text length increases, the detection accuracy of RoBERTa-MoD also improves, which is consistent with the discussion in our main paper.

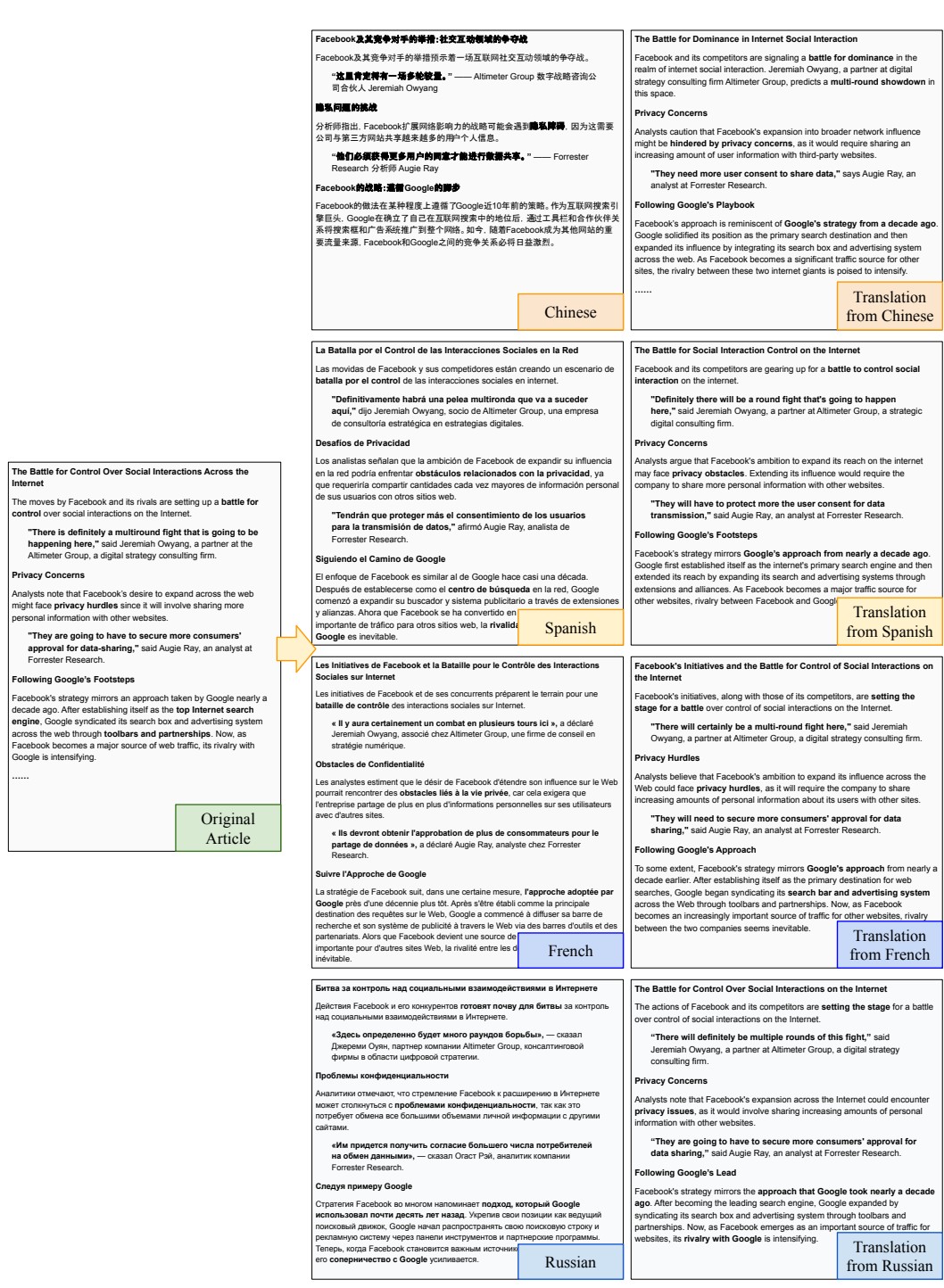

Figure 6: **Round-trip Strategy for Article Translation.** This strategy allows us to automatically produce translated articles from existing datasets, eliminating the need for additional data collection. See Appendix C for discussion.

Table 4: Fine-grained MGT Detection with Different Input Lengths. See Appendix D for discussion.

| Model Scale | Llama3 -8B | Qwen1.5 -7B | StableLM2 -12B | ChatGLM3 -6B | Qwen2.5 -7B | **avg mAP** | **AUROC** |
|---|---|---|---|---|---|---|---|
| mAP on VisualNews (Liu et al., 2021) | | | | | | | |
| Length=64 | 69.45 | 76.52 | 66.61 | 68.01 | 66.99 | 69.89 | 86.51 |
| Length=128 | 73.04 | 81.67 | 71.88 | 72.44 | 71.75 | 74.16 | 89.72 |
| Length=256 | 75.25 | 83.09 | 70.53 | 72.96 | 73.06 | 75.48 | 90.23 |
| Length=512 | 81.05 | 82.99 | 77.23 | 79.25 | 78.10 | 79.84 | 91.96 |

