# OpenReview forum: "Fine-Grained Machine-Generated Text Detection"
_ICLR.cc/2025/Conference — Submitted to ICLR 2025_

### Official Review · Reviewer_6eQi · 2024-11-01

**Soundness:** 3
**Presentation:** 2
**Contribution:** 2
**Rating:** 5
**Confidence:** 3

**Summary:**

The paper investigates fine-grained MGT detection; the authors propose to categorize an input text into four classes: human-written, machine-generated, machine-paraphrased, and machine-translated. They introduce a data preparation process to generate articles across different fine-grained categories, enabling the automatic creation of training and evaluation data for the task. The paper introduces a RoBERTa-based Mixture of Detectors (RoBERTa-MoD) for fine-grained MGT detection, which leverages multiple domain-optimized detectors for more robust and generalized performance. The paper presents theoretical proof that the method outperforms a single detector, and experimental findings show an improvement in mAP over prior work on six various datasets: GoodNews, VisualNews, WikiText, Essay, WP, and Reuters.

**Strengths:**

— The idea of separating into three categories is intuitive and transferred from the actual use cases. However, it makes the task more complicated as the generation style is similar for paraphrasing/translating/generation using the same models.

— It is beneficial for nature to theoretically assert if some models and approaches are worse or better without extensive model training.

— Reproducibility statements, limitations, and ethical considerations are included. Clear contribution. The code and data will be publicly released upon acceptance.

**Weaknesses:**

Some questions for reproducibility of the research.

— Please include information about the strategy of round-trip translation and paraphrasing. The rationale behind round-trip translation is not discussed. Was the translation performed for each language using all the models?
Are the authors certain that the prompt "Paraphrase/Translate the following article: x." was used consistently across all models? If so, how does LLAMA determine which language to translate? What are the specific languages used?
How do authors gather paraphrases and translations? For each article, there should be a minimum of four different translations or paraphrases. Have the authors confirmed that removing the first sentence from the text does not alter the meaning of the translation?

— For the training data, have the authors saved the model's proportion distribution? No statistics for the training corpora. How many examples were used for each class?
Section 3.1 needs more details.

— Line 372  `All methods were fine-tuned on data from Llama-3 (Touvron et al., 2023) and Qwen-1.5 (Bai et al., 2023) and then evaluated on all LLMs.`
Which data? How much data in which format? Maybe the authors mean: "evaluated on the data from all the LLMs"? I guess the methodology was to check in out-of-domain evaluation so that the data formed with different models and not evaluated by the same models in an LM-as_judge manner. The formulations are confusing. It is not clear from the texts what data you trained detectors, fine-tuned which exact models, and with what models you evaluated what.

`Llama-3 and Qwen-1.5 are in-domain generators for training the detector, and StableLM-2, ChatGLM-3, and Qwen-2.5 are out-of-domain generators to evaluate the model’s generalization ability.`
Am I right that, based on the data from models StableLM-2, ChatGLM-3, and Qwen-2.5, were the detectors not trained?

`However, in the same dataset, human-written articles in the training and testing sets may follow similar data distributions`
There is no information on whether they may or may not.

— Line 325: `LLM-DetectAIve is directly trained on fine-grained MGT data, which can be considered as a fine-tuned RoBERTa.`
Why should it be considered? No explanations/justifications

— Figure 5. It seems like the Qualitative Results are based on one example. Paraphrasing can change factual information as well as translation. The paper needs some quantitative metrics to catch it.

**Questions:**

— The translated and paraphrased texts, if created using LLMs, can also contain misinformation and factual errors. It depends on the LLM; the percentage of errors is much rarer than that of machine generation, but it still needs to be checked.

— "As discussed in the Introduction," add the link to the Introduction section.

— Line 307: Set the spaces in cite like in:  `GoodNews(B`

— The paper would improve from the footnotes to direct links for the open datasets and a clear explanation of the steps with data processing.

— Table 1 would improve if the information about the model's size was added.

— It's a bit strange not to see the results section.

— The results would be interesting to check on the different lengths of the output. We could see the correlation with length.

— Factual error: LLM-DetectAIve distinguishes four categories: (i) human-written, (ii) machine-generated, (iii) machine-written, then machine-humanized, and (iv) human-written, then machine-polished. The idea is different from paraphrasing.

---

> ### Author Response · Authors · 2024-11-24
>
> We thank the reviewer for their valuable comments, we appreciate their time and will use their suggestions to improve our paper. We note that our primary contribution is to our task fine-grained text detection, which has not been previously explored. This contribution itself is notable, and makes our paper valuable as we show that prior work finds our task challenging, especially for differentiating paraphrased and translated text, our new categories.
>
>
> >Was the translation performed for each language using all models?
>
> Yes
>
> > What are the specific languages used?
>
> As noted on L179 these are: Chinese, Spanish, Russian, and French
>
> > How do authors gather paraphrases and translations? For each article, there should be a minimum of four different translations or paraphrases
>
> We translate and paraphrase each article. Thus, for the 10K GoodNews articles we use for training, there would be 40K translations (4 languages * 10K = 40K), 10K paraphrased articles, and 10K machine generated articles.
>
> > For the training data, have the authors saved the model's proportion distribution? No statistics for the training corpora. How many examples were used for each class? Section 3.1 needs more details.
>
> For training statistics like the number of examples used for each class, please refer to Section 4.1. E.g., L308 states that 10K GoodNews articles were used for training.  These were then used to generate our categories.
>
> >Have the authors confirmed that removing the first sentence from the text does not alter the meaning of the translation?
>
> As noted on L178, we input the entire article to the generator.  While L185 states that for Llama-3 paraphrasing we remove the second paragraph, we note this is due to the fact Llama-3 typically responds with answers that begin with “Here is the polished version:” which would tell the detector that this is paraphrased text.  Also, to reiterate, this was only done for Llama-3
>
> >*Llama-3 and Qwen-1.5 are in-domain generators for training the detector, and StableLM-2, ChatGLM-3, and Qwen-2.5 are out-of-domain generators to evaluate the model’s generalization ability.* Am I right that, based on the data from models StableLM-2, ChatGLM-3, and Qwen-2.5, were the detectors not trained?
>
> We did not train on StableLM-2, ChatGLM-3, and Qwen-2.5.  Instead, they are used for evaluation to demonstrate that our approach generalizes.
>
> > The translated and paraphrased texts, if created using LLMs, can also contain misinformation and factual errors. It depends on the LLM; the percentage of errors is much rarer than that of machine generation, but it still needs to be checked.
>
> We agree that it is unlikely, but not impossible that the translated and paraphrased texts also contain misinformation. That exactly explains why fine-grained MGT detection is needed. By fine-grained MGT detection, we can mark that machine-generated texts are mostly contain misinformation, machine-translated and paraphrased texts are less likely to contain misinformation, and distinguish them from human-written texts, which can not be done by the traditional MGT detection task.
>
> >However, in the same dataset, human-written articles in the training and testing sets may follow similar data distributions There is no information on whether they may or may not.
>
> This has been well established in prior work, including a discussion by the dataset authors of GoodNews and VisualNews.  In fact, the motivation to collect news articles from different sources in VisualNews is motivated exactly by this fact (as well as noting the years in which the articles were collected, as there can be a shift in topics over time).  This is also shown in generated text detection work, e,g.,
>
> Zhongping Zhang, Wenda Qin, and Bryan A. Plummer. Machine-generated text localization. In Findings of the Annual Meeting of the Association for Computational Linguistics: ACL, 2024a.
> >Line 325: LLM-DetectAIve is directly trained on fine-grained MGT data, which can be considered as a fine-tuned RoBERTa. Why should it be considered? No explanations/justifications
>
> LLM-DetectAIve (Abassy et al., 2024) applies a RoBERTa-single model trained on fine-grained MGT data, and, thus, can be considered a fine-tuned RoBERTa.
>
> > It's a bit strange not to see the results section.
>
> The **Experiments** section already includes the results, which are split between our fine-grained MGT task (Section 4.3), Zero-shot Fine-grained detection (Section 4.4), and traditional MGT detection (Section 4.5).

---

> ### Author Response · Authors · 2024-11-24
>
> > Factual error: LLM-DetectAIve distinguishes four categories: (i) human-written, (ii) machine-generated, (iii) machine-written, then machine-humanized, and (iv) human-written, then machine-polished. The idea is different from paraphrasing.
>
> Human-written and machine-generated are the standard categories and human-written then machine-polished is similar to paraphrasing as it is based on human text and then adjusted by a machine.  Machine-written and machine-humanized are all generated, and, thus, could be considered (a perhaps more challenging) machine generated text.  Thus, from these broad definitions our statement is accurate, but we will provide a more nuanced discussion to include this perspective.  That said, as we note in our paper, LLM-DetectAIve is concurrent work, i.e., it is not prior work.

---

> ### Author Response · Authors · 2024-11-24
>
> > Table 1 would improve if the information about the model's size was added.
>
> We have added the information about the model’s size (See Table 1&2) in the paper, where they range in size from 7B-12B parameters.
>
> > The results would be interesting to check on the different lengths of the output. We could see the correlation with length.
>
> We have reported the results of different lengths using our RoBERTa-MOD model in Appendix D. Similar to prior work, smaller text tend to perform worse.  We see a relatively small drop in AP going from 128-256 length inputs, but the raw scores even for shorter text is still relatively good (nearly 70 avg mAP)

---

> > ### Author Response · Authors · 2024-12-02
> >
> > Hello Reviewer 6eQi- As we are coming to the last roughly 24 hours or so in the discussion period we were hoping you would review our rebuttal and consider improving your score. Thank you for your efforts!

---

### Official Review · Reviewer_CWcV · 2024-11-03

**Soundness:** 2
**Presentation:** 3
**Contribution:** 2
**Rating:** 3
**Confidence:** 3

**Summary:**

This paper addressing the detecting of machine generated text proposed to extend the traditional binary classification (machine or human generated) to a four-classes problem: human-authored, machine-generated, machine-translated or machine-paraphrased. The authors propose a multi-class classification model uses 4 Roberta-models and a gating mechanism. The whole network is trained in two stages

**Strengths:**

* An interesting twist to an important problem

* An analysis that combines both theoretical insights and empirical ablations

**Weaknesses:**

* Motivation: the idea behind a 4-class problem (instead of the more traditional 2-class) is motivated by the fact that machine-modified text (translated or corrected) might not be as harmful as text that is generated by a model from scratch. While this is a somehow appealing explanation, it would be interesting to see if that difference actually has an empirical impact. Sect 4.5 makes this attempts, by applying the proposed method on existing benchmarks, obtaining good scores (although not always outperforming the current state-of-the-art). More interesting to me would be an analysis on how `Binoculars` performs on the new datasets (considered as binary problem). Another interesting aspect would be to verify if the more fine-grained classification actually results in a better binary classification.

**Questions:**

* Could you address the weakness mentioned above?

---

> ### Author Response · Authors · 2024-11-24
>
> We thank the reviewer for their valuable comments, we appreciate their time and will use their suggestions to improve our paper. We note that our primary contribution is to our task fine-grained text detection, which has not been previously explored. This contribution itself is notable, and makes our paper valuable as we show that prior work finds our task challenging, especially for differentiating paraphrased and translated text, our new categories.
>
> > How Binoculars performs on the new datasets (considered as binary problem)?
>
> We compared our method with Binoculars in the binary classification task in Table 3. The experimental results show that Roberta-MoD achieves comparable performance to Binoculars. We would also like to point out that our paper focuses on the fine-grained classification task, and Binoculars cannot be applied to this task. As mentioned in Section 2 and Section 4.1, since fine-grained categories in MGT are also generated by LLMs, theoretically, machine-translated and machine-paraphrased text would be classified as machine-generated text based on the statistical features extracted by these methods.  I.e., our goal is not to improve the binary classification task as it is sufficient when considering in-the-wild data that can include these fine-grained categories.
>
> > If the more fine-grained classification actually results in a better binary classification?
>
> We reported the performance of Roberta-MoD in binary classification in Table 3. The experimental results show that compared to single detectors, RoBERTa-MoD not only performs better on fine-grained classification but also on binary classification.

---

> > ### Comment · Reviewer_CWcV · 2024-11-25
> >
> > Thank you for those additional comments.
> >
> > I had seen Sect 4.5 previously, and with it the fact that the new methods does not always improve performance as measured in the old settings. Paying closer attention now I see that Reuters is clear outlier, where your method gets +15p in F1. How do you explain that jump? Is it in Precision or Recall?
> >
> > The new dataset that you introduced could be converted into the binary case by for example merging machine-generated, paraphrased and translated into one big bucket. My question was how previous methods perform on that binary tasks.
> >
> > You did not address my main comment on the _motivation_ of this work. If a new setting is introduced, its motivation should either be self-explanatory, inspired by documented real cases or proof that by looking at this problem through a new lens previous method can be improved (on their playground, eg - binary classification here).
> > While the first of these reasons is subjective, it is my opinion that none of these conditions are uphold in this paper

---

> ### Author Response · Authors · 2024-11-25
>
> >You did not address my main comment on the motivation of this work. If a new setting is introduced, its motivation should either be self-explanatory, inspired by documented real cases or proof that by looking at this problem through a new lens previous method can be improved (on their playground, eg - binary classification here). While the first of these reasons is subjective, it is my opinion that none of these conditions are uphold in this paper
>
> Apologizes, as your comment seemed to rehash part of our motivation, we did not understand you wanted us to comment on it directly.  To address this question, we have, in fact, discussed generated text detection with people who wish to deploy these models as opposed to machine learning researchers, which is what lead us to working on this task.  To help enlighten you on some of the challenges with prior work, let us consider an example in content moderation (e.g., X, Weibo, Facebook, Reddit, etc).  One challenge these websites face is due to bots that automatically generate content that can flood user's news feeds or forms.  Generated text detectors could help spot these bots, but if a user had used a LLM to translate or paraphrase their post, they could also get flagged by the generated text detector.  This could be resolved by having a human content moderator review their posts, but this can be costly.  Removing the posts is also not ideal since it diminishes the user experience, which may cause them to have their posts reinstated (requiring content moderators to review) or have them stop using the site altogether, reducing revenue.  In contrast, using a detector from our work, the content moderator could ignore the paraphrased and translated text, and thereby be able to target the bots as opposed to the real users.  The concurrent work of Abassy et al. 2024 also highlights that the binary classification setting is simply not sufficient for many applications.  As such, the importance of fine-grained generated text detection tasks is not subjective, but rather, a necessary evolution for many applications of these models.
>
> >I had seen Sect 4.5 previously, and with it the fact that the new methods does not always improve performance as measured in the old settings.
>
> While our approach does not improve performance over every dataset, it does improve performance in all settings over the individual detectors we use to compose our RoBERTa-MoD approach.  While Binoculars does outperform our approach in some settings, as we note on L504, since this model uses a metric-based approach using a fixed threshold to identify generated text, it does not generalize to fine-grained generated text classification.  However, given that RoBERTa- MoD does improve performance over individual models, if we wanted to focus only on the generated text detection task we could simply include Binoculars as one of the detectors in our model.  As discussed next, this should result in a model that works better across more text sources.
>
> >Paying closer attention now I see that Reuters is clear outlier, where your method gets +15p in F1. How do you explain that jump? Is it in Precision or Recall?
>
> To clarify, Table 3 reports only a 1 point improvement over RoBERTa-MPU on Reuters, and we use a RoBERTa model as one of our detectors.  Thus, the difference stems from the fact that the metric-based Binoculars does not generalize to this setting compared with a learned detector like RoBERTa-MPU.

---

> ### Author Response · Authors · 2024-12-02
>
> Hello Reviewer CWcV- As we are coming to the last roughly 24 hours or so in the discussion period we were hoping you would review our rebuttal and consider improving your score.  We have responded to your follow-up questions and hope these resolve your questions. Thank you for your efforts!

---

### Official Review · Reviewer_WW4c · 2024-11-04

**Soundness:** 2
**Presentation:** 3
**Contribution:** 2
**Rating:** 5
**Confidence:** 4

**Summary:**

The paper presents an in-depth study on fine-grained Machine-Generated Text (MGT) detection, which can classify text into four categories: human-written, machine-generated, machine-paraphrased, and machine-translated. The authors note that existing detectors struggle with out-of-domain text, particularly for translated or paraphrased text. To address this, they propose a RoBERTa-based Mixture of Detectors (RoBERTa-MoD) which uses multiple detectors optimized for different domains to improve performance. The authors provide a theoretical proof that their method outperforms a single detector and experiments show a 5-9% improvement in mean Average Precision (mAP) over previous work on six diverse datasets. The authors also introduce a data preparation process to generate articles across different fine-grained categories, enabling automatic creation of training and evaluation data for the task.

**Strengths:**

1) It presents an in-depth study on fine-grained Machine-Generated Text (MGT) detection, a topic that has been overlooked in previous studies. By classifying text into four categories (human-written, machine-generated, machine-paraphrased, and machine-translated), the research contributes significantly to the field.
2) The paper introduces the RoBERTa-based Mixture of Detectors (RoBERTa-MoD), a novel method that uses multiple domain-optimized detectors for more robust and generalized performance. This method addresses the performance drop on out-of-domain texts, a key challenge in MGT detection.
3) The paper's quality is evident in the theoretical proof provided for the method's superiority over a single detector. Additionally, the research is significant as it achieved a 5-9% improvement in mean Average Precision (mAP) over prior work on six diverse datasets.

**Weaknesses:**

1) The concept of employing a mixture of experts (MoE) for a task is not unusual, considering its extensive application in various tasks such as general LLM, summarization, and machine translation. While the application of MoE in text detection is relatively new, it is not a groundbreaking concept in terms of its fundamental idea.
2) The study lacks an ablation analysis on MoD. Questions such as the influence of the number of detectors on the results, the effect of different corpus (domains) on detection, and the likelihood of the router specializing to specific detectors given an input article, remain unanswered.
3) The paper doesn't delve deep into the confusion between classes, for instance, between translate and human/paraphrase. As the paper is centered on the fine-grained detection of machine-generated text, such analyses about the challenges of fine-grained detection are anticipated and would offer valuable insights to the readers.
4) The experimental results may not have been compared in a fair manner. However, due to the lack of clear descriptions of the settings for the baselines, I will reserve my judgment until the rebuttal period, during which I expect a response.

**Questions:**

LN053: what is the special of your method compared to Abassy et al., 2024 given that they both do fine-grained MGT detection?

LN091: have you considered GPT models for the detection? What is the underlying reason for choosing RoBERTa?

LN320: how do you calculate the AUROC for the 4-class classification problem?

Table 1: the experiments are limited to generations from open-source models. Have you considered latest closed-source models like GPT4, Gemini, and Claude3?

Table 1: In my view, a RoBERTa-single model trained on the same data is a demanded baseline to demonstrate the advantage of the MoE design.

Table 1: Are models like ChatGPT-D, RoBERTa-MPU, and LLM-DetectAIve trained on the same data as the proposed method?

LN427: How are your zero-shot experiments designed? When we say zero-shot, we refer to a method without any task specific training. How could your MoD method do the zero-shot given that it requires a joint training of the router and the detectors?

---

> ### Author Response · Authors · 2024-11-24
>
> We thank the reviewer for their valuable comments, we appreciate their time and will use their suggestions to improve our paper. We note that our primary contribution is to our task fine-grained text detection, which has not been previously explored. This contribution itself is notable, and makes our paper valuable as we show that prior work finds our task challenging, especially for differentiating paraphrased and translated text, our new categories.
>
> > LN053: what is the special of your method compared to Abassy et al., 2024 given that they both do fine-grained MGT detection?
>
> LLM-DetectAIve (Abassy et al., 2024) applies a RoBERTa-single model trained on fine-grained MGT data. Therefore, as we mentioned in the caption of Table **1**, we can consider LLM-DetectAIve as a fine-tuned RoBERTa. In addition, Abassy et al. does not consider categories like translated text.  That said, as we note, this paper is unpublished and was made available on arxiv months after the deadline for ICLR 2025 to consider it concurrent work.  Thus, it should not be considered “prior work” with significant comparisons made to it.
>
>
> > LN091: have you considered GPT models for the detection? What is the underlying reason for choosing RoBERTa?
>
> Yes, we performed relevant experiments in Table 3. As we discussed in Section **4.1**, the GhostBuster data contains ChatGPT, ChatGPT-turbo, ChatGLM, GPT4all, Claude, and StableLM. Since most existing detectors (e.g., OpenAI-Detector, ChatGPT-Detector, RoBERTa-MPU) apply the RoBERTa structure, we choose RoBERTa as our backbone to make fair comparisons.
>
> > LN320: how do you calculate the AUROC for the 4-class classification problem?
>
> Following GhostBuster (Verma et al., 2024), we calculated the AUROC score using the “roc_auc_score” function in scikit-learn, which can be used for multiclass classification. More details can be found in their official document:
> https://scikit-learn.org/1.5/modules/generated/sklearn.metrics.roc_auc_score.html .
>
>
> > Table 1: the experiments are limited to generations from open-source models. Have you considered latest closed-source models like GPT4, Gemini, and Claude3?
>
> We choose open-source models for fine-grained categories generation. For latest closed-source models like Claude, ChatGPT, we reported the experimental results in Table **3**. As we discussed in Section **4.1**, this datasets contains *ChatGPT, ChatGPT-turbo, ChatGLM, GPT4all, Claude, and StableLM*.
>
>
> > Table 1: In my view, a RoBERTa-single model trained on the same data is a demanded baseline to demonstrate the advantage of the MoE design.
>
> LLM-DetectAIve (Abassy et al., 2024) applies a RoBERTa-single model trained on fine-grained MGT data. Therefore, as we mentioned in the caption of Table 1, we can consider LLM-DetectAIve as a fine-tuned RoBERTa.This also answers your first question.
>
>
> >Table 1: Are models like ChatGPT-D, RoBERTa-MPU, and LLM-DetectAIve trained on the same data as the proposed method?
>
> Yes, as we mentioned in Section 4.3: “All methods were fine-tuned on data from Llama-3 (Touvron et al., 2023) and Qwen-1.5 (Bai et al., 2023), and then evaluated on all LLMs”.
>
>
> > LN427: How are your zero-shot experiments designed? When we say zero-shot, we refer to a method without any task specific training. How could your MoD method do the zero-shot given that it requires a joint training of the router and the detectors?
>
> Zero-shot experiments means we trained all methods on GoodNews, and evaluated them on VisualNews and WikiText. That said, the router and the detectors are trained only on GoodNews articles, which is the same as all other baselines, and then directly evaluated on VisualNews and WikiText articles.

---

> > ### Comment · Reviewer_WW4c · 2024-11-25
> > **Thanks for clarification**
> >
> > I appreciate your explanation regarding the questions, particularly the part where you clarified that the zero-shot experiments are essentially OOD experiments due to the training on GoodNews and testing on VisualNews and WikiText. However, I would prefer to maintain my existing score, as the main weaknesses have not been resolved.

---

> ### Author Response · Authors · 2024-11-25
>
> Thank you for your response.   We would like to know what weaknesses exactly you are referring to (i.e., what can we do to help resolve your questions to improve your score)?
>
> In particular, we note that you did not argue against our task, which is the first to explore fine-grained generated text detection.  This task address a critical shortcoming of prior work, namely that detecting whether a piece of text is generated is not sufficient for many applications, as paraphrased and translated text can often be considered a more benign use of these models.  As this is the first proposal of this task, simply doing an analysis that adapts existing approaches is often deemed sufficient for publication as the main contribution is the task.  The concurrent work of Abassy et al. also suggests that this task is important and worth studying.  However, most of your review centered on the MoD, which both is not our main contribution and you did note is a new application and the fairness of the results (the latter of which you seem to acknowledge is resolved).  As such, we would like to discuss your justification in more detail so we can improve our work.

---

> > ### Author Response · Authors · 2024-12-02
> >
> > Hello Reviewer WW4c- As we are coming to the last roughly 24 hours or so in the discussion period we were hoping you would review our rebuttal and consider improving your score.  In particular, as we noted in our last response, if you could highlight exactly what weaknesses you feel are not well addressed, especially in light of the fact that our main contribution (as highlighted by the title of our paper) is the task we aim to address. Thank you for your efforts!

---

### Official Review · Reviewer_N6JH · 2024-11-08

**Soundness:** 2
**Presentation:** 2
**Contribution:** 3
**Rating:** 5
**Confidence:** 3

**Summary:**

The paper presents novel task of fine-grained MTD detection, where the detector should be able to predict 4 labels: humn-generated, machine-generated, machine-translated and machine-paraphrased. Novel architecture for this task is roposed, refered as MoD (Mixture of Detectors), consisting several detectors for individual domains, and the trainable router.

Theoretical results are presented demonstrating the theoretical benefits of the proposed architecture over the individual detector.

The evaluation is done on several popular MTD datasets for various generator models. Besides, the OOD evaluation is presented. The method outperformes  the most of the considered baselines by a large margin

**Strengths:**

- The paper proposes novel important tasks of fine-grained MTD. The presented analysis demonstrates that indeed Machine-Generated, Machine-Translated and Machine-Paraphrased texts have unique features and different usage scenarios, and it is important to distinguish them
- Novel architecture of MTD is proposed
- The evaluation is done of large amount of data domains and generator models, and includes OOD setup. The proposed method outperforms most of the considered baselines by a large margin.

**Weaknesses:**

- The theoretical framework in Sec 3.3 is not quite clear (see Questions)

- The benefit of the proposed method over standard Mixture-of-Expert is not clear

- In section 3.1, the statistics of the generated data is absent (see Q4)

- The most of the baseline models are Roberta-based classifiers. Comparing to the proposed method, they have k times less parameters, where k is the number of individual detectors. Only two baselines do not belong to this class: RoBERTa-MoS and Binoculars; comparing to them, the proposed method have marginal to no improvement

- No comparison to Roberta-MoS in OOD setup

- In Sec. 4.3, the Qualitative result paragraph describes the properties of the dataset rather then the classifier results. There is no information about the typical errors, or any other qualitative description of the results of the proposed detector.

**Questions:**

Q1. Please explain the notation used in the Definition and Theorems in Sec. 3.3
- What is Patch? Does it correspond to a set of tokens, or a subset of features (e.g. coordinates) in the text embeddings, or smth else?
- What is feature vector $v_k$? The index indicates the whole cluster, but it is used for the description of the individual data point
- What does the notation $y\alpha v_k$ mean? Is it a vector multiplied by 2 scalars $y$ and $\alpha$ ?

In general, could you please provide the example of the considered setup in the Machine-Generated Text Detector domain, defining patches, clusters, data features, distraction features and the noise.

Q2. How many detectors were used for each dataset? Does this number correspond to the theoretical bound from Theorem 2?

Q3. How are ChatGPT-D and Roberta-MPU atapted to fine-grained MTD setup? Are they fine-tuned on the same dataset as MoD?

Q4. Please describe the statistics of the train/test dataset used in Tables 1 - 3 (its fine-grained part). For Generator models, please specify which version of each model is used (i.e. number of parameters)

---

> ### Author Response · Authors · 2024-11-24
>
> We thank the reviewer for their valuable comments, we appreciate their time and will use their suggestions to improve our paper. We note that our primary contribution is to our task fine-grained text detection, which has not been previously explored. This contribution itself is notable, and makes our paper valuable as we show that prior work finds our task challenging, especially for differentiating paraphrased and translated text, our new categories.
>
> > Q1.1. What is Patch? Does it correspond to a set of tokens, or a subset of features (e.g. coordinates) in the text embeddings, or something else?
>
> Following Chen et al. (2022), a patch represents a subset of the input, where each subset exhibits features corresponding to different attributes. For example, the first patch $x^{(1)}$ may present features related to the target text cluster(*e.g.*, the input text is a news article, then $k$ corresponds to the news domain), the second patch $x^{(2)}$ may capture features corresponding to other categories, and the third patch may contain noise features. To simplify, a patch can be considered as a subset of features from the input text.
>
> > Q1.2. What is feature vector $v_k$? The index indicates the whole cluster, but it is used for the description of the individual data point
>
> Following Chen et al. (2022), $v_k$ is a label signal vector that presents the features of cluster $k$ (*e.g.*, the news domain in Q1.1). Although $v_k$ primarily provides a cluster-level representation, it can be used to describe the individual data points by indicating the relationship between the data point and its associated cluster.
>
> > Q1.3. What does the notation $y\alpha v_k$ mean? Is it a vector multiplied by 2 scalars $y$ and $\alpha$ ?
>
> $y$ represents the ground truth label, $\alpha$ is a scalar. The notation $y \alpha v_k$ denotes a patch belonging to cluster $k$ that exhibits the signal of the ground truth label $y$. To simplify, a patch given by $y\alpha v_k$ should be classified as the text category $y$ and domain $k$.
>
> > Q1.4. In general, could you please provide the example of the considered setup in the Machine-Generated Text Detector domain, defining patches, clusters, data features, distraction features and the noise.
>
> **Patch:**
> Given a piece of the input text $\{w_1, w_2, w_3, … , w_n\}$, a patch can be a subset of the input. For instance, tokens $\{w_1, w_5, w_6, w_9, …, w_n\}$ contain features indicative of the machine-generated category (*Data Features*). Tokens $\{w_2, w_3, w_7,..., w_{n-1}\}$ contain features corresponding to irrelevant categories (*Distracting Features*). Tokens $\{w_4, w_8,...,w_{n-2}\}$ contain noisy features.
> **Cluster:**
> Clusters represent groups of texts with similar feature distributions. For example, in our task, cluster $v_1$ can be the GoodNews domain, cluster $v_2$ can be the VisualNews domain, and cluster $v_3$ can be the Wikipedia domain.
>
> **Data Features:**
> Data features are the characteristics derived from the text input that are relevant for the detection task. For example, given a piece of machine-paraphrased text, it should be correctly classified according to its data features.
>
> **Distraction Features:**
> Distraction features are irrelevant or confounding features that may exist in the data but do not directly contribute to identifying machine-generated text. For example, given a piece of machine-paraphrased text, its distraction features can exhibit the machine-generated or human-written signals.
>
> **Noise:**
> Noise includes random or unpredictable variations in the data that obscure meaningful patterns.
>
> > Q2. How many detectors were used for each dataset? Does this number correspond to the theoretical bound from Theorem 2?
>
> Three detectors were used for each dataset. Since $M$ is greater than 2 in Theorem 2, this number satisfies the requirement.
>
>
> > Q3. How are ChatGPT-D and Roberta-MPU atapted to fine-grained MTD setup? Are they fine-tuned on the same dataset as MoD?
>
> As discussed in Section 4.3, we fine-tuned all baselines using the same dataset as MoD. Specifically, we fine-tuned the classifier heads of ChatGPT-D and RoBERTa-MPU on the training data of Llama-3 and Qwen-1.5, enabling them to be applied to fine-grained MGT detection.
>
> > Q4. Please describe the statistics of the train/test dataset used in Tables 1 - 3 (its fine-grained part).
>
> We provide statistics on the number of samples for each split at the start of Section 4.1.  For example, as we noted on L308 for GoodNews we use 10K randomly selected articles and 2K each for testing/validation, with the same amount used to evaluate on VisualNews (note that we do not train using VisualNews as it is used as an out-of-distribution experiments).

---

> ### Author Response · Authors · 2024-11-24
>
> >Q4 (cont). For Generator models, please specify which version of each model is used (i.e. number of parameters)
>
> We have added this information to Tables 1&2, where the models range from 7B-12B parameters.

---

> > ### Author Response · Authors · 2024-12-02
> >
> > Hello Reviewer N6JH- As we are coming to the last roughly 24 hours or so in the discussion period we were hoping you would review our rebuttal and consider improving your score. Thank you for your efforts!

---

### Meta-Review · Area_Chair_maHX · 2024-12-18

**Metareview:**

This paper proposes a new multi-class task formulation for machine text detection by subcategorizing machine text into generated, translated and paraphrased text. A RoBERTA-based mixture of domains model with a router mechanism is used for the task. Experiments are presented across a variety of domains.

**Strengths:** The paper poses an interesting question about machine text, by claiming that not all machine text may be the same. They present results across different domains and compare with multiple baseline approaches.

**Weaknesses:** The authors claim that their main contribution is the task - but I’m not convinced that model translated and model paraphrased text might be in principle different from machine generated text. The assumption about factuality seems somewhat strong: humans also make many false claims! Some empirical evidence of this would make for a more compelling case for this paper. Secondly, it is unclear how their RoBERTa-MoD classifier does on the OOD task when adapted to a binary setting (also pointed out by reviewer CWCv).

**Reason for decision**: See above. The main contribution of the work: the four-way classification task seems somewhat arbitrarily defined; reviewers were not satisfied with the depth of analysis and experimentation.

**Additional Comments On Reviewer Discussion:**

Reviewers pointed multiple issues with the method, which the authors’ response could not fully justify. There seem to be several issues with the depth of analysis of the model itself (lack of satisfactory ablations) as well as the results. The task (labels) itself might also be somewhat arbitrarily defined. While additional experiments and further arguments were presented, it seemed that the authors might have missed the main crux of reviewers’ concerns. Reviewers perhaps did not feel compelled to engage with a long discussion with the authors due to this reason.

---

### Decision · Program_Chairs · 2025-01-22

Reject